# Double Pulse LIBS Analysis of Metallic Coatings of Fusionistic Interest: Depth Profiling and Semi-Quantitative Elemental Composition by Applying the Calibration Free Technique

**Salvatore Almaviva *** , **Francesco Colao, Ivano Menicucci and Marco Pistilli**

ENEA, Italian National Agency for New Technologies, Energy and Sustainable Economic Development, Frascati Research Center, via Enrico Fermi, 45, I-00040 Frascati, Italy
* Correspondence: salvatore.almaviva@enea.it

**Abstract:** In this work we report the characterization of thin metallic coatings of interest for nuclear fusion technology through the ns double-pulse LIBS technique. The coatings, composed of a tungsten (W) or tungsten-tantalum (W-Ta) mixture were enriched with deuterium (D), to simulate plasma-facing materials (PFMs) or components (PFCs) of the next generation devices contaminated with nuclear fuel in the divertor area of the vacuum vessel (VV), with special attention to ITER, whose divertor will be made of W. The double pulse LIBS technique allowed for the detection of D and Ta at low concentrations, with a single laser shot and an average ablation rate of about 110 nm. The calibration free (CF-LIBS) procedure provided a semi-quantitative estimation of the retained deuterium in the coatings, without the need of reference samples. The presented results demonstrate that LIBS is an eligible diagnostic tool to characterize PFCs with high sensitivity and accuracy, being minimally destructive on the samples, without PFCs manipulation. The CF-LIBS procedure can be used for the search for any other materials in the VV without any preliminary reference samples.

**Keywords:** LIBS; ITER divertor; plasma-facing-components; tungsten

## 1. Introduction

The International Thermonuclear Experimental Reactor (ITER) [1] is currently under construction in Cadarache, south of France, and is designed to demonstrate the feasibility of energy production from nuclear fusion. In the ITER VV, beryllium (Be) will be used as the PFM of the first wall [2], while tungsten will be used as main material of PFCs of its divertor [2,3].

Compared to the present fusion devices, the ITER will be operated with a higher number of deuterium-tritium (D-T) plasma discharges and longer pulse durations, therefore the first wall and divertor PFCs must withstand higher harsh radiation, intense particles fluxes and heat loads, with increased material erosion, dust formation and T retention, thus leading to safety issues [3–6].

The above-mentioned processes, particularly T retention, need to be carefully and continuously monitored by a dedicated suite of diagnostics, possibly operating in situ, sampling larger areas of the VV without removing PFCs from their positions.

Among the suitable diagnostics, laser-based methods are able to accomplish these tasks [7], and can probe the PFCs of the first wall and the divertor without sample manipulation. Laser-based methods include LIDS (Laser Induced Desorption Spectroscopy) [8], LIBS (Laser Induced Breakdown Spectroscopy) [9] and LID-MS (Laser Induced Desorption combined with Mass Spectroscopy) [10].

LIBS has already demonstrated its versatility for these tasks, having been used and tested in many of the currently operating tokamaks in two different configurations: as a diagnostic tool external to the tokamak, delivering the laser pulses on the PFCs of the VV through optical windows [11], and as a compact and lightweight diagnostic, suitable to

be mounted on a robotic arm and carrying out the analyses directly inside the tokamak, during its shutdown or maintenance periods [12]. In LIBS [13], one or more short laser pulses of high energy (duration $\leq$ ns [14]) are released on the sample, be it solid, liquid or gaseous [15], inducing the breakdown of the sample and plasma formation (typical values T $\approx$ 10,000 K, electron density $\approx 10^{15}$–$10^{17}$ cm$^3$), containing neutral atoms, ions, molecular fragments and free electrons. The excited chemical species recombine by emitting electromagnetic radiation [13–16]. The spectral analysis of this radiation allows for the identification of the species present in the plasma with the intensity of emission lines generally proportional to the concentration of those particular species [17,18]. In the double-pulse LIBS (DP-LIBS), two subsequent laser pulses are delivered on the samples with a short inter-pulse delay, enhancing the intensity of the LIBS signal compared to a standard single-pulse LIBS [19,20].

The most important advantages of LIBS in the field of PFCs characterization are that it does not require any sample pretreatment, it offers simultaneous multi-element detection, it allows isotopes detection, it is quasi non-destructive of the sample (few μg of sample are needed), and it has depth profiling capabilities and multi-elemental stratigraphy. Moreover, LIBS measurements can be carried out both in a vacuum or at high pressure. Finally, it can provide quantitative and semi-quantitative estimation of the species present in the plasma. For these reasons, LIBS will represent a valid complement to the analyses currently carried out on the PFCs of the current fusion tokamaks for those of the next generation, which will present more stringent limits on handling and the ex situ measurement of PFCs.

In this work, we report on a DP-LIBS analysis characterizing fusion-relevant metallic coatings composed of W and W-Ta mixed materials contaminated with D. A high depth resolution, down to 110 nm per single laser shot, was obtained, still preserving a good SNR of the spectral features ascribable to the emitting species, so that all the elements in the coatings were clearly identified through their spectral lines. A discussion on the more relevant spectral features of these and other elements relevant for ITER near the spectral region of the Balmer alpha emission of hydrogen isotopes is presented. A semi-quantitative estimation of the deuterium contained in the samples is performed by applying the calibration free (CF-LIBS) method [17,21,22] that does not need any reference or calibration sample.

## 2. Materials and Methods

The samples have dimensions of 10 mm $\times$ 12 mm $\times$ 1 mm and have been produced by the National Institute for Laser, Plasma, and Radiation Physics (NILPRP), Romania, by high power impulse magnetron sputtering (HiPIMS) [23] in a high vacuum chamber pumped down to a typical base pressure lower than $10^{-4}$ Pa. The sample's composition and thickness was characterized by applying Glow Discharge Optical Emission Spectroscopy (GDOES) [24].

Four samples were measured, each with different coatings in composition and thickness:

(1)  The first one is composed of mixed tungsten-nitrogen-deuterium coating, W~80%, N~14%, D~6% atomic concentration, 4.5 μm thick, (W80/N14/D6 in the following);

(2)  The second one is still composed of W-N-D but with a different concentration; W~70%, N~26%, D~4%, 6.5 μm thick, (W70/N26/D4 in the following);

(3)  The third one is composed of a mixed tungsten-tantalum-deuterium coating, W~90%, Ta~3.5%, D~4–6.5%, and a thickness of 4.5 μm, (W90/Ta3.5/D5 in the following);

(4)  Finally, the fourth one is composed by W-Ta-D with W~70–80%, Ta~4.5%, D~6–15%, and a thickness 2.8 μm, (W75/Ta4.5/D10 in the following).

For all samples, the superficial coating was deposited on Mo substrates. The first two samples aimed to simulate the ITER PFCs of the divertor region contaminated with nuclear fuel, whereas the last two aimed to simulate fuel-contaminated PFCs made by W-Ta alloys, which seems to exhibit better properties than W in terms of erosion and hydrogen loading, as was observed in dedicated experiments with e-beams in JUDITH [2,3] and in GLADIS [2] facilities.

The DP-LIBS system is fully described in [12]; here we briefly summarize that it was composed by a Montfort M-nano PIV double pulse laser, operating at $\lambda$ = 1064 nm with a pulse duration = 8 ns, with both beams focused orthogonally on the sample surface. The inter-pulse delay between the two pulses was set to 65 ns, and the energy to 58–59 mJ for both. The pulses were focused on the sample surface with a 1" plano-convex lens of 50 mm focal length. Each laser-induced spot was measured by an optical microscope and had a diameter of ~300 μm on the samples.

The LIBS plasma was collected by a 2", 200 mm focal length lens and focused on a circular fiber bundle (Ø = 1.094 mm) composed of 17 fibers (Ø = 0.245 mm each), each 6 m in length. At the exit, the fibers are arranged in a linear bundle 4.12 mm long in order to fit the entrance slit of a TRIAX 550 ISA JOBIN-IVON monochromator, equipped with a 2400 g/mm diffraction grating, able to record with a high resolution (~0.1 Å at 500 nm), and a spectral region of about 8–10 nm. The signal acquisition was performed by an ANDOR Istar DH320T-18F-63 ICCD with an 18 mm intensifier and a minimum gating time of 10 ns.

The spectrometer's calibration was performed by looking at the wavelength position of the two Ne I emission lines at 653.29 nm and 659.9 nm of a low pressure Ne lamp and applying a quadratic function to the considered range to account for the observed shift.

The gate delay of the ICCD was set to 4.25 μs and a gate width of 1 μs; these parameters were fixed after optimization, aiming to reduce the interference of the two $D_\alpha/H_\alpha$ nearby emission lines of D and H at 656.1 nm and 656.28 nm (see the following section) in air, where D comes from the sample and H from the environmental hydrogen. Indeed, these two emission lines are the most intense of H and D in the visible spectral region but suffer from a remarkable Stark broadening [25] that make it difficult to resolve them from the whole spectral line-shape if a careful optimization of the acquisition parameters is not performed.

The elemental composition and the stratigraphic structure of the coatings have been determined by recording the emission intensity of the W, D, H and Ta lines as a function of the applied laser shots on the same point of the sample. The spectral region studied was between 653 and 660 nm, comprising not only the $D_\alpha/H_\alpha$ emissions but also evident emission lines of W and Ta, as shown in Table 1 according to the NIST database [26]. These emission lines were used to clearly detect the transition from the superficial layer to the Mo substrate.

**Table 1.** Relevant emission lines of W-Ta-D in the 653–660 spectral region according to [26].

| Samples | Element | Wavelength in Air (nm) |
|---|---|---|
| All | W | 653.29 |
| All | W | 653.81 |
| All | D | 656.1 |
| W90/Ta3.5/D5 W75/Ta4.5/D10 | Ta | 656.16 |
| All | H | 656.28 |
| All | W | 656.32 |
| W90/Ta3.5/D5 W75/Ta4.5/D10 | Ta | 656.42 |
| All | W | 657.39 |
| W90/Ta3.5/D5 W75/Ta4.5/D10 | Ta | 657.48 |

Sequences of 50–60 shots were applied to the samples in order to study the whole transition from the superficial coatings to the Mo substrate.

The CF-LIBS method was applied to carry out a semi-quantitative estimation of the D content of the samples compared to W, which was assumed to be the reference material of the coatings, simulating the composition of the divertor in ITER [1]. CF-LIBS is widely

used for different types of metals and alloys, including those of interest with regard to fusion [27] and also for non-metallic samples, such as soils [28] or rocks [29]. The method is based on the assumption that (1) materials ablated from the sample and emitting in the plasma have the same stoichiometry of the sample; (2) the plasma is optically thin; and (3) the plasma is in Local Thermodynamic Equilibrium (LTE) [17].

In this case, the line integrated intensity of the transition between two levels $E_k$ and $E_i$ of an atomic species can be expressed as [17]:

$$I_{ki} = C_s A_{ki} \frac{g_{ki} e^{-\frac{(E_{ki})}{K_B T}}}{U_s(T)} \tag{1}$$

where $C_s$ is the concentration of the species (atomic %), $A_{ki}$ is the transition ($k \rightarrow i$) probability for the given line, $g_{ki}$ is the k-level degeneracy, $E_{ki}$ is the excited level energy, $K_B$ is the Boltzmann constant, $T$ is the plasma temperature and $U_s(T)$ is the partition function of the emitting species at temperature $T$. Starting from the experimental intensities, one can obtain the Boltzmann plot (BP) [30] of each species, whose slope gives the electron temperature of the species and whose intercept is proportional to the concentration of that species in the plasma [17].

## 3. Results

### 3.1. Depth Profiling

The samples W80/N14/D6 and W75/Ta4.5/D10 were chosen to highlight the depth profiling capabilities of the DP-LIBS system (Figure 1). Two sequences of the applied laser shots are shown in Figure 1, where the transition of the LIBS spectrum from the superficial layer to the Mo substrate is clearly observed:

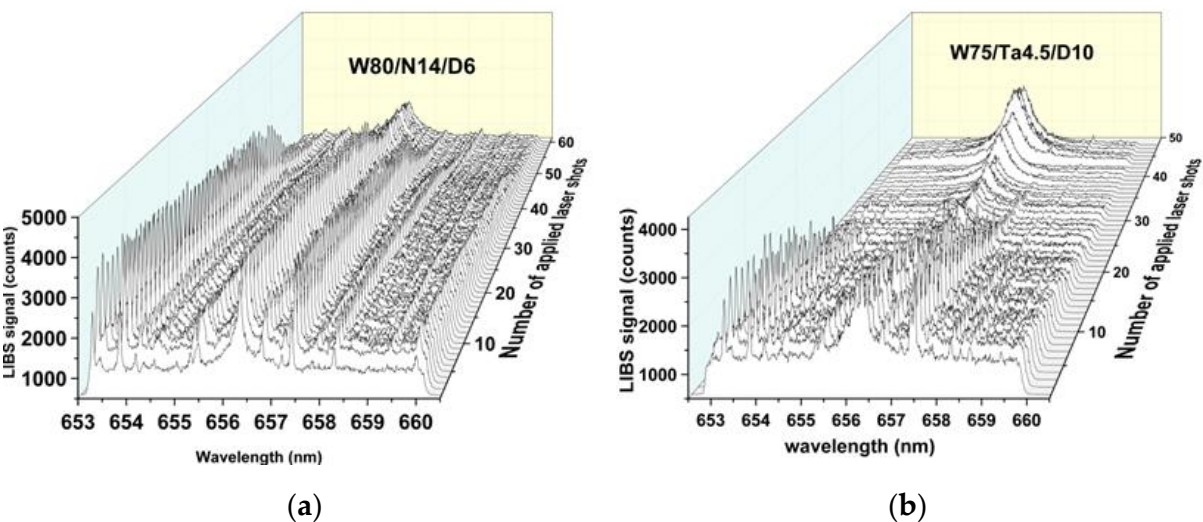

(a)                                                                                        (b)

**Figure 1.** (**a**) sequence of laser shots applied on the W80/N14/D6 sample and (**b**) on the W75/Ta4.5/D10 sample in the spectral region 653–660 nm.

In these spectra, the W-Ta-H-D lines appear in the first shots ablating the superficial coating, and start to decay when the Mo substrate is reached. The Mo substrate, on the contrary, does not exhibit evident emission lines in this spectral range that can alter the intensities of the lines of the superficial elements, except for the strong $H_\alpha$ emission line of hydrogen, the species that heavily contaminates the substrate.

To better highlight the spectral emissions of the elements of the coating, the second to the eleventh shots were averaged for each sample and considered representative of the surface layer. The first laser shot was eliminated to avoid the contribution of any surface contamination, and the average was stopped at only ten successive shots to also avoid any decrease in the spectral signal due to the complete ablation of the surface layer. The average

spectrum for each sample is shown in Figure 2, together with the spectral assignments for each emission line:

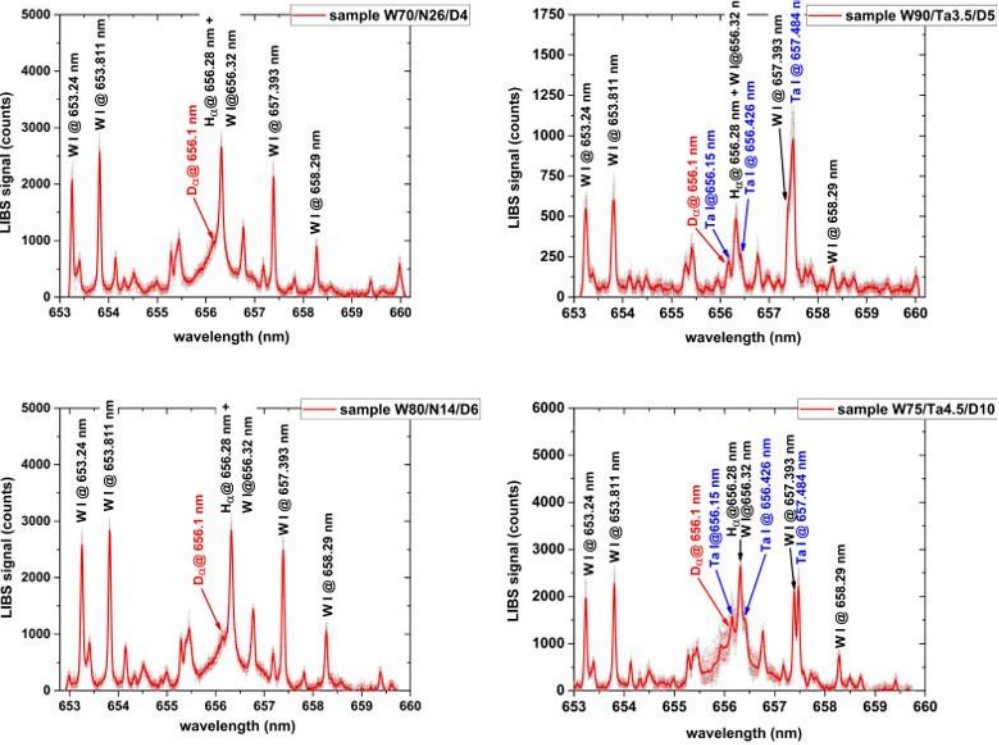

**Figure 2.** LIBS spectra (in light gray color) of the four samples W70/N26/D4 sample (**up**) and W80/N14/D6 (**bottom**) samples. The red spectrum is the average spectrum of the second to eleventh laser shot. Light red bars represent the SD related to this spectrum. Emission lines of W, Ta, D and H are labeled in different colors (black for W and H, red for D, and blue fot Ta).

To estimate the number of laser shots needed to completely ablate the surface layer in all of the samples, the intensities of some W, Ta, and D spectral lines reported in Table 1 were monitored and compared with the GDOES profiles of the coatings provided by NILPRP. The results for samples W80/N14/D6 and W75/Ta4.5/D10 are shown in Figure 3.

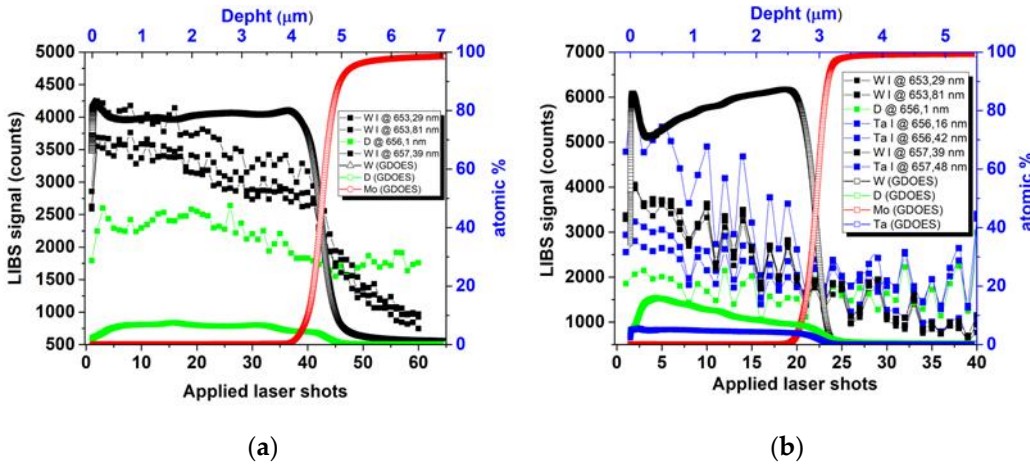

|           |           |
|-----------|-----------|
| (**a**)   | (**b**)   |

**Figure 3.** (**a**) Comparison of the LIBS intensities (filled squares) and GDOES profiles (open squares) of the W80/N14/D6 sample and (**b**) the W75/Ta4.5/D10 sample. The vertical yellow lines mark the nominal thickness of the superficial layer to the substrate.

The transition from the surface layer to the substrate is well observed by looking at the intensity variation of the W and Ta lines as a function of the applied laser shots. The

intensities tend to decrease abruptly when the superficial layer is completely ablated. For the $D_\alpha$ line this transition to the substrate is less evident. This effect is due to the strong contamination of the Mo substrate with H, increasing the intensity of the nearby $H_\alpha$ peak. This peak, being Stark-broadened, interferes with the residual signal at 656.1 nm, in the position of the $D_\alpha$ peak, making it difficult to follow the intensity of the real $D_\alpha$ peak from the superficial layer to the substrate.

The average ablation rate per single laser shot was evaluated by the following formula:

$$AAR = \frac{d_{\mu m}}{N_{laser\ shots}} \tag{2}$$

where $d_{\mu m}$ is the GDOES thickness (μm) of the superficial layer and $N_{laser\ shots}$ is the number of laser shots required to reach the substrate.

For the two W80/N14/D6 and W75/Ta4.5/D10 samples, formula (1) gave an $AAR_{W80/N14/D6} = 0.11$ μm and $AAR_{W75/Ta4.5/D10} = 0.12$ μm.

The use of laser sources with even shorter pulse lasers (e.g., ps or fs lasers) will allow for better ablation properties both for reduced $AAR$ per pulse and thermal effects [27].

### 3.2. Calibration Free Semi-Quantitative Analysis

For each sample, the average spectrum of ten laser shots applied on the coating in the spectral region 653–660 nm was considered as the reference spectrum (Figure 2), showing some evident W I emission lines free from interference, in addition to the $D_\alpha$ emission line.

### 3.2.1. Electron Temperature

The electron temperature was evaluated using the common form of the BP, given as follows:

$$\ln\left(\frac{I_{ki}}{g_k A_{ki}}\right) = \frac{-E_k}{K_B T} + \ln\left(\frac{C_s F}{U_s(T)}\right) \tag{3}$$

with the same notations as in Formula (3) and $F$ a constant depending on the experimental conditions. Measurement of the intensities of atomic lines originating from different excited states of the same species allows for the evaluation of plasma temperature, provided that the transition probabilities and statistical weights of these lines are known. By plotting the left-hand side of Equation (3) vs. $E_k$ and fitting the data with a line, the $1/K_B T$ gives the electron temperature $T$. Assuming a plasma largely dominated by tungsten ions, the electron temperature was evaluated through the BP of tungsten lines in the spectral region between 461 and 471 nm. In Figure 4, the data for the sample W80/N14/D6 gives $T = 9800 \pm 400$ K, with the error being estimated as the standard deviation (SD) of the slope of the linear fit.

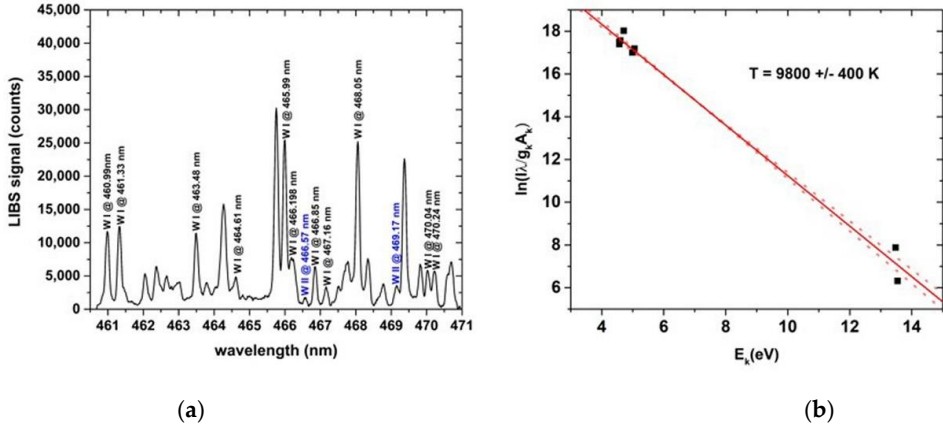

(**a**)         (**b**)

**Figure 4.** (**a**) LIBS spectrum of sample W80/N14/D6 in the spectral region between 461–471 nm. Labeled in black atomic emission lines and in blue ionic emission lines. (**b**) BP of the W emission lines between 461 and 471 nm; the red line represents the best linear fit, while the red-dotted lines represent linear fits with maximum and minimum slope within the SD.

### 3.2.2. Electron Density

The electron density of the plasma was evaluated through the line broadening of the $H_\alpha$ lines at 656.28 nm in the Mo substrate. These lines have been fitted as a Voigt function centered at 656.28 nm from the experimental line-shape in the region of 655.5–656.5 nm, together with other components present in the spectrum. In Figure 5, the spectral line-shape for each sample and the corresponding components are shown.

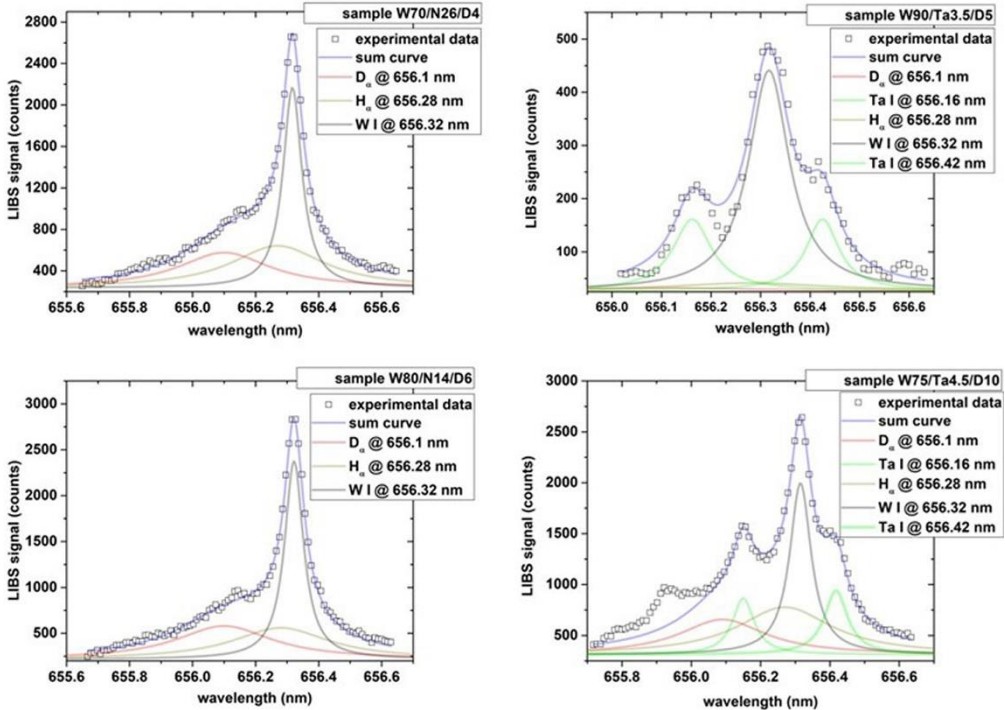

**Figure 5.** Fits of the experimental line-shapes of the four samples in the $D_\alpha/H_\alpha$ spectral region. Each spectral component of W, Ta, D, H and their sum curve is shown in different colors.

According to Gigosos [31], the full width at half area (*FWHA*) of these $H_\alpha$ lines are related to the plasma electron density through the following expression:

$$FWHA(\text{nm}) = 0.549 \left( \frac{N_e}{10^{23}\text{m}^{-3}} \right)^{0.67965} \tag{4}$$

The *FWHA* of the four $H_\alpha$ peaks have been computed and added to Formula (4) to provide the value of the electron densities reported in Table 2:

**Table 2.** Electron density estimated form the *FWHA* of the $H_\alpha$ peaks.

| Sample | Electron Density ($\times 10^{16}$ cm$^{-3}$) |
|---|---|
| W70/N26/D4 | 5.16 |
| W80/N14/D6 | 5.27 |
| W90/Ta3.5/D5 | 5.27 |
| W75/Ta4.5/D10 | 4.52 |

### 3.2.3. Plasma Optically Thin

To ensure that the plasma is optically thin, i.e., that the radiation emitted by an excited atom is not re-absorbed by another atom in a lower energy state, it is necessary to check if self-absorption of the emission lines occurs. This can be obtained from the intensity ratio of two lines of the same chemical species, free of interference, with a procedure similar to that reported in [32]. For these particular cases we compared the theoretical intensity ratio

of the non-resonant, non-interfering W I lines at 653.24 nm and 653.811 nm with the ones obtained experimentally and reported in Table 3.

**Table 3.** Experimental vs. theoretical intensity ratio of W non-resonant lines.

| Samples | Exp. Intensity Ratio W(653.24)/W(653.811) | Theor. Intensity Ratio W(653.24)/W(653.811) |
|---|---|---|
| Sample W80/N14/D6 | 0.91 | 0.953 |
| Sample W70/N26/D4 | 0.88 | 0.953 |
| Sample W90/Ta3.5/D5 | 0.93 | 0.953 |
| Sample W75/Ta4.5/D10 | 0.86 | 0.953 |

The consistency between the experimental and theoretical intensity ratio is within an absolute error $\leq$ 10% for all cases, confirming an optically thin LIBS plasma.

### 3.2.4. Local Thermodynamic Equilibrium

To check the LTE condition, it is necessary that the LIBS plasma is characterized by an electron density and temperature sufficient to ensure a high collision rate, whose lower limit for LTE is given by the so-called McWhirter criterion [33]:

$$N_e \geq 1.6 \cdot 10^{12} T^{1/2} \Delta E^3 \tag{5}$$

where $N_e$ (cm$^3$) is the electron density, $T$ (K) is the plasma temperature and $\Delta E$ (eV) is the largest energy transition for which the condition holds. In the case of W, $\Delta E$ = 4.48 eV and $N_e$(W) $\geq$ 1.42·10$^{16}$ cm$^3$. In the case of Ta, $\Delta E$ = 2.38 eV and $N_e$(Ta) $\geq$ 2.14·10$^{15}$ cm$^3$. In the case of D (and H) $\Delta E$ = 10.2 eV and $N_e$(D) $\geq$ 1.68·10$^{17}$ cm$^3$. The McWhirter criterion is fulfilled for W and Ta, while for H and D, the condition is only partially fulfilled, so it is expected that only a partial LTE (pLTE) is present for H and D in the plasma and the population of the 2 s level from the transition 1 s–> 2 s ($\Delta E$ = 10.2 eV) does not follow the Saha–Boltzmann distributions.

### 3.2.5. Evaluation of the Deuterium Retained in the Coatings

The D concentration ratio was evaluated by the intensity ratio of the integrated D$_\alpha$ and W emission lines in the coatings. The D$_\alpha$ line was obtained by fitting the spectral region around 655.5 and 656.5 nm, as shown in Figure 5. The instrumental function of the system was fitted by a Lorentzian function with FWHM of $4.5 \times 10^{-2}$ nm, measured by evaluating the FWHM of the two Ne I emission at 653.29 nm and 659.9 nm of a low pressure Ne lamp. Therefore, the experimental emission lines were fitted with a Lorentzian function by using the nonlinear Curve Fitting package of Origin 2016 (OriginLab).

The considered W lines were 653.24 nm and 653.811 nm, free of interference, whose atomic parameters are known in the literature.

Finally, the intensity ratio was rescaled by the W percentage in the coatings, with a procedure similar to that applied in [34].

Table 4 shows the experimental and theoretical [D]/[W] ratios obtained by applying the procedure.

**Table 4.** Semi-quantitative estimation of the D concentration in the four analyzed samples, compared to the nominal concentration given by the GDOES analysis.

| Sample | Nominal [D/W] (%) | Exp. [D/W] (%) |
|---|---|---|
| W70/N26/D4 | 7.5 | 7.06 |
| W80/N14/D6 | 5.7 | 7.78 |
| W90/Ta3.5/D5 | 5.6 | 0.42 |
| W75/Ta4.5/D10 | 13 | 8.02 |

## 4. Discussion

The *AAR* obtained for the two sets of coatings showed to be comparable and in agreement with the relative low content of Ta in the W-Ta mixtures. It was possible to clearly detect the LIBS spectrum with a high SNR and carry out depth profiling with high axial sensitivity using the DP-LIBS technique, recording the LIBS spectrum from a single laser shot. It is well known that the use of even shorter pulse lasers (e.g., ps or fs lasers) allows for better ablation properties, both in terms of reduced *AAR* per pulse and thermal effects that can alter the stoichiometry of the sample [27]. In any case, these lasers currently have relatively lower single-pulse energies than ns lasers and, typically, more laser pulses are released at each analysis point to have LIBS spectra with sufficient SNR [27], increasing the *AAR* and reducing the advantages of these ultra-short pulse sources. With the advent of ultra-short pulse lasers of greater power, both laser ablation without thermal effects and single-pulse measurements with reduced *AAR* should be readily available.

The semi-quantitative estimation of the D retained in the coatings from this study is compatible with the nominal elemental concentrations, although some relevant deviations are present, especially in the cases of the coatings with Ta. Indeed, in the case of the alloys with Ta, the deconvolution of the experimental spectrum by taking into account its theoretical components is particularly complex, given the presence of multiple spectral emissions in a narrow spectral region, penalizing a reliable evaluation of the intensities of the two Balmer emission lines of D and H, which were found to be evidently underestimated in the case of sample W90/Ta3.5/D5. For the samples without Ta, the deconvolution of the experimental signal appeared simpler, and the semi-quantitative estimate of D more precise, with only the contribution of the interfering W I line at 656.32 nm, apart from the two Balmer alpha peaks, being present in this spectral region. In the case of a real sample from the divertor region of ITER, recent analyses performed on the Joint European Torus (JET) with an ITER-like wall suggests a composition of the superficial deposits largely dominated by Be [35]. For these compositions a LIBS spectrum with the Be II emission line at 655.84 nm and a Be I emission line at 656.45 [26] is expected, not or weakly interfering with the spectral window of the $D_\alpha$/$H_\alpha$ peaks.

**Author Contributions:** Conceptualization, S.A.; methodology, S.A. and F.C., formal analysis, S.A.; investigation, I.M. and M.P.; writing—original draft preparation, S.A. All authors have read and agreed to the published version of the manuscript.

**Funding:** This work has been carried out within the framework of the EUROfusion Consortium and has received funding from the Euratom research and training programme 2014–2018 and 2019–2020 under Grant No. 633053. The views and opinions expressed herein do not necessarily reflect those of the European Commission.

**Data Availability Statement:** Data can be available upon request from the authors.

**Acknowledgments:** The authors also wish to thank Eduard Grigore for providing details about the composition of the samples, and Violeta Lazic for the fruitful discussions, design and support in assembling the device.

**Conflicts of Interest:** The authors declare that they have no conflict of interest.

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
