# Peer review of "Double Pulse LIBS Analysis of Metallic Coatings of Fusionistic Interest: Depth Profiling and Semi-Quantitative Elemental Composition by Applying the Calibration Free Technique"

_jne, doi:10.3390/jne4010015_

Round 1

Reviewer 1 Report

GENERAL

·         Acronyms should be spelled out also after the abstract, whenever they are firstly used in the main body of the manuscript.

SPECIFIC

·         Line 3 and line 145:  „Depht“ -> „Depth“

·         Line 9: “….fusionistic interest….”doesn’t exists as an expression. Please rephrase it.

·         Line 50: Is the Word “species” the right one to describe all the previous mentioned material (matter) fragments?

·         Line 74: Some paragraphs need to start with the first letter indent

·         Line 155: “substarte”

·         Line 156: Word missing after “spectral”

·         Line 259: “…thermodYnamic…”

·         Line 280: The authors mentioned that the data were fitted but being more precise this is a spectrum deconvolution procedure and the authors need to describe a bit more or reference the method they used cause it is the main calculation to determine the promised concentration ratio of D.

Author Response

We thank the reviewer for his comments and observations and we apoligize for some typos and grammatical errors that may have distracted him from reading the manuscript.  Following detailed responses to his observations:

 Line 3 and line 145: „Depht“ -> „Depth“

Corrected

Line 259: “…thermodYnamic…”

Corrected

Line 9: “….fusionistic interest….”doesn’t exists as an expression. Please rephrase it

Changed in: "of interest for nuclear fusion technology"

Line 50: Is the Word “species” the right one to describe all the previous mentioned material (matter) fragments?

We understand the doubt of the reviewer; "species" relate to the ablated chemical species, not the produced fragments. According to the reviewer's suggestion we changed the next sentence in the text to better clarify this point;  from "These species..." in "The excited chemical species..."

Line 74: Some paragraphs need to start with the first letter indent

Done

 Line 155: “substarte”

Corrected

Line 156: Word missing after “spectral”

Corrected in :"spectral range"

Line 280: The authors mentioned that the data were fitted but being more precise this is a spectrum deconvolution procedure and the authors need to describe a bit more or reference the method they used cause it is the main calculation to determine the promised concentration ratio of D.

According to the reviewer's comment we have better explained the adopted fitting procedure and we referred to reference [31] for further details. The text in the manuscript was changed as follows:

"The instrumental function of the system was fitted by a Lorentzian function with FWHM ­of 4.5 × 10−2 nm, measured by evaluating the FWHM of the two Ne I emission at 653.29 nm and 659.9 nm of a low pressure Ne lamp. Therefore, the experimental emission lines were fitted with a Lorentzian function by using the nonlinear Curve Fitting package of Origin 2016 (OriginLab) and applying a procedure similar to that reported in [31].

Reviewer 2 Report

In the following paper, entitled, “Double pulse LIBS analysis of metallic coatings of fusionistic interest: depht profiling and semi-quantitative elemental composition by applying the calibration free technique” the Almaviva et al, reported the characterization of thin metallic coatings of fusionistic interest through the ns double-pulse libs technique. The results are interesting. I would recommend the manuscript to be accepted after major revision. The following should be noted before acceptance:

1.      The novelty statement needs to address in the introduction section.

2.      The author needs to include the schematics of experimental setup.

3.      The authors need the address the further details about spectrometer’s calibration.

4.      In Fig. 4(b), the error bars should be included. In addition, the detail about uncertainties ±400 K is missing in text. From where these uncertainties originates?

5.      How the author measured the laser spot (Page. 3, line 100).

6.      How laser energy was optimized. (In page. 3 line 97)

7.      In Eq. 1, please specify the representation of indexed i and k; and why author did not incorporated the wavelength in Eq. 1.

8.      In Figure. 2, deconvolute the Dα line for central wavelength.

9.      Please specify Figure 1 in text (Page 4; section 3, lien 154)

10.  In Figure 2, why author did not identified some emission lines around 654-656 nm?

11.  The explanation of Figure 3 in text is insufficient.

12.  In Table. 2 dots should be removed. Furthermore, incorrect unit of number density was used (replace cm3 to cm-3).

13.  Further details of CF-LIBS should be included and cite the following papers:

https://doi.org/10.3390/molecules27123754

https://doi.org/10.3390/molecules27155048

14.  How CF-LIBS limitation were cross checked with another standard spectroscopic techniques such as XRF, EDX etc.

15.  Cite following papers on DP-LIBS in introduction on DP-LIBS;

https://doi.org/10.1134/S0030400X21040068

16.  Image quality should be improved throughout the manuscript as it’s difficult to read.

Author Response

We thank the reviewer for his observations and we report in the following detailed responses and corrections made on the text.

1. The novelty statement needs to address in the introduction section.

Done, we have added in the introduction further statements about the novelty of the LIBS technique by adding the following sentences: (now in page 1, line 63-65)

"...for these reasons, LIBS will represent a valid complement to the analyzes currently carried out on the PFCs of the current fusion tokamaks for those of the next generation which will present more stringent limits on handling and ex-situ measurement of PFCs"

2. The author needs to include the schematics of experimental setup

According to the reviewer's suggestion we remind to ref [12] in the text for the schematics in order to avoid multiple repetitions of the same setup used in different publications. To do so we added the following sentence (now in line 99)

"The DP-LIBS system was fully described in [12]; here we briefly summarize that it was composed by..."

3. The authors need the address the further details about spectrometer’s calibration:

According to the reviewer's suggestion we have added the following sentence in the manuscript: (now in lines 112-114):

"The spectrometer’s calibration was performed by looking at the wavelength position of the two Ne I emission lines at 653.29 nm and 659.9 nm of a low pressure Ne lamp and applying a quadratic function to the considered range to account for the observed shift"

4. In Fig. 4(b), the error bars should be included. In addition, the detail about uncertainties ±400 K is missing in text. From where these uncertainties originates?

The coordinates of the points in the Boltzann plot are taken:

Y axis (ordinate): from a function including spectral parameters of the considered  atomic transition: These values are known from literature and are multiplied by the experimental line integral. This experimental integral is obtained from the integral of the Lorentzian fit of each curve and therefore is a parameter whose error is not provided.

X axis (abscissa) is obtained from literature values of the excitation energies of the upper energy levels of the considered transitions whose error is not known. The reported error on the temperature value is obtained from the standard error on the slope of the linear fit and has been reported on the figure, in agreement with the referee's suggestion. In the new figure the error was reported as two fitting lines having the maximum and the minimum value of the slope within the standard error. The sentence immediately before figure 4 in the text has been changed in: “…giving T = 9800 ± 400 K, with the error being estimated as standard deviation (SD) of the slope of the linear fit” and the figure caption as: “…the red line represent the best linear fit, the red-dotted lines represent linear fits with maximum and minimum slope within the SD.”

5. How the author measured the laser spot (Page. 3, line 100).

The laser spot was measured by using an optical microscope. This was reported now in the text according to the reviewer's suggestion: (now page line 103-104): "Each laser-induced spot was measured by an optical microscope and had a diameter of ~ 300 μm on the samples"

6.      How laser energy was optimized. (In page. 3 line 97)

Laser energy and, more in general, all the acquisition parameters (interpulse delay, gate width, gate delay, etc) have been optimized to better resolve the two nearby emission lines of Dα and Hα of Deuterium and Hydrogen, still preserving an acceptable SNR. This is pointed out in the text in the next sentences after lines 97. (page 3 line 116 -123):

"...Gate delay of the ICCD was set to 4.25 ms, gate width 1 ms; these parameters were fixed after optimization aiming to reduce the interference of the two Dα/Hα nearby emission lines of D and H at 656.1 nm and 656.28 nm (see the following section) in air, where D comes from the sample and H from the environmental Hydrogen. Indeed, these two emission lines are the most intense of H and D in the visible spectral region but suffer from a remarkable Stark broadening [25] that make difficult to resolve them from the whole spectral line-shape if a careful optimization of the acquisition parameters is not performed"

In Eq. 1, please specify the representation of indexed i and k; and why author did not incorporated the wavelength in Eq. 1.

Done. We have included both indexes i ad k of the transition k→ i in formula (1) and better specified the indexes in the text (now page 3 line 145): "...where Cs is the concentration of the species (atomic %), Aki is the transition (→ i)..."

8. In Figure. 2, deconvolute the Dα line for central wavelength.

The deconvolution of all the emission lines of figure 2 close to Dα  are reported in figure 5

9. Please specify Figure 1 in text (Page 4; section 3, line 154).

Done: added at the end of the specified sentence

10. In Figure 2, why author did not identified some emission lines around 654-656 nm?

We have omitted to identify these lines because relatively weak, sometimes interfering, not functional to the clarity of the spectrum and for the next figure 2 showing the intensity behaviour.  

11. The explanation of Figure 3 in text is insufficient.

According to the reviewer's suggestion we have changed the explanation of figure 3 adding more details as follows (page 6, lines 191-198): 

"...The transition from the surface layer to the substrate is well observed by looking at the intensity variation of the W and Ta lines as a function of the applied laser shots. The intensities tend to decrease abruptly when the superficial layer is completely ablated. For the Dα line this transition to the substrate is less evident. This effect is due to the strong contamination of the Mo substrate with H, increasing the intensity of the nearby Hα peak. This peak, being Stark-broadened  interferes with the residual signal at 656.1 nm, in the position of the Da peak, making it difficult to follow the intensity of the real Da peak from the superficial layer to the substrate."

12 In Table. 2 dots should be removed. Furthermore, incorrect unit of number density was used (replace cm3 to cm-3).

Done

13. Further details of CF-LIBS should be included and cite the following papers:

https://doi.org/10.3390/molecules27123754

https://doi.org/10.3390/molecules27155048

Done

14 How CF-LIBS limitation were cross checked with another standard spectroscopic techniques such as XRF, EDX etc.

According to the reviewer's suggestion and reported in the text in table 4, we have cross-checked the CF-LIBS results with the nominal concentration provided by GDOES. Our analysis is restricted to the quantification of Deuterium. XRF and EDX cannot check the Deuterium or Hydrogen content of the sample, so we compared our results with the GDOES data only. 

15. Cite following papers on DP-LIBS in introduction on DP-LIBS;

https://doi.org/10.1134/S0030400X21040068

Done

16.  Image quality should be improved throughout the manuscript as it’s difficult to read

According to the reviewer's suggestion we have improved the image quality, in partiular with larger labels.   

Round 2

Reviewer 2 Report

The authors addressed all concerns in a satisfactory manner.